# Uncovering the genetic diversity and adaptability of Butuo Black Sheep through whole-genome re-sequencing

**Zengwen Huang[1], Jing Wang[1], Dongming Qi[1], Xiaoyan Li[1], Jinkang Wang[2], Jingwen Zhou[3], Yan Ruan[2], Youse Laer[2], Zhangjia Baqian[3], Chaoyun Yang[1]\***

**1** Xichang University, Xichang, China, **2** Butuo County Agriculture and Rural Affairs Bureau, Xichang, China, **3** Butuo County Forestry and Grassland Bureau, Xichang, China

\* chaoyuny@yeah.net

**Data Availability Statement:** The raw data was preserved at the NCBI database (accession number. PRJNA1079389).

## Abstract

The Butuo Black Sheep (BBS) is well-known for its ability to thrive at high altitudes, resist diseases, and produce premium-quality meat. Nonetheless, there is insufficient data regarding its genetic diversity and population-specific Single nucleotide polymorphisms (SNPs). This paper centers on the genetic diversity of (BBS). The investigation conducted a whole-genome resequencing of 33 BBS individuals to recognize distinct SNPs exclusive to BBS. The inquiry utilized bioinformatic analysis to identify and explain SNPs and pinpoint crucial mutation sites. The findings reveal that reproductive-related genes (*GHR*, *FSHR*, *PGR*, *BMPR1B*, *FST*, *ESR1*), lipid-related genes (*PPARGC1A*, *STAT6*, *DGAT1*, *ACACA*, *LPL*), and protein-related genes (*CSN2*, *LALBA*, *CSN1S1*, *CSN1S2*) were identified as hub genes. Functional enrichment analysis showed that genes associated with reproduction, immunity, inflammation, hypoxia, PI3K-Akt, and AMPK signaling pathways were present. This research suggests that the unique ability of BBS to adapt to low oxygen levels in the plateau environment may be owing to mutations in a variety of genes. This study provides valuable insights into the genetic makeup of BBS and its potential implications for breeding and conservation efforts. The genes and SPNs identified in this study could serve as molecular markers for BBS.

## Introduction

Sheep (*Ovis aries*) is one of the most vital livestock species globally, supplying meat, milk, wool, and other by-products for human consumption and use. Sheep have been domesticated for over 10,000 years, dating back to Southwest Asia [1]. Since then, sheep have been widely distributed across various regions and climates, adapting to different environmental and human-induced pressures. As a result, sheep have developed noteworthy diversity in phenotypic traits, including coat color, horn shape, tail length, wool quality, and disease resistance [2–4]. Coat color is an easily recognizable and distinctive phenotypic trait of sheep. The production of melanin pigment by melanocytes in the skin and hair follicles determines the coat

**Funding:** We are supported by Xichang University Doctoral Start-up Funding (No. 117281746) and Butuo Black Sheep Data File (No. 117281599).

**Competing interests:** The authors have declared that no competing interests exist.

color in sheep. Eumelanin (black or brown) or pheomelanin (red or yellow) is produced depending on the activity of enzymes and receptors involved in melanin synthesis and transport [3, 5–7]. The genetic basis of coat color variation in sheep has been the subject of extensive study, revealing the roles of several genes, such as *MC1R* [8], *ASIP* [9, 10], *TYRP1* [7, 11], *MITF* [12, 13], and *KIT* [14].

The Yi people in Butuo County, China, maintain a strong connection with sheep, treating them as more than just a source of food and clothing but also a symbol of their identity and social status. The Butuo Black Sheep (BBS) has garnered attention due to its adaptability, disease resistance, and ability to yield high-quality wool [2]. Furthermore, BBS is recognized as a valuable genetic resource for sheep breeding and conservation purposes [15]. The distinguishing features of BBS include its black coat color, large body size, long tail, and coarse wool. However, limited information is available on BBS's genetic diversity and origin, particularly the genes accountable for their black coat color.

Next-generation sequencing (NGS) technologies have revolutionized genomics research, enabling high-throughput and cost-effective sequencing of whole genomes [16, 17]. As a result, the identification and characterization of SNPs in various species, including sheep [2, 18], is now comprehensive. Among individuals of the same species, SNPs are the most common type of genetic variation, and they have a crucial role in genetic diversity, disease susceptibility, and response to environmental factors [19]. Identifying and analyzing SNPs can provide valuable insights into the genetic makeup of a species and identify potential genetic markers associated with economically important traits [18]. Several recent studies have utilized whole-genome resequencing to investigate genetic diversity and identify SNPs in sheep breeds [20, 21]. For example, a survey of Australian Merino sheep identified over 15 million SNPs, providing insights into the genetic basis of wool quality [20]. In a previous study, several SNPs were revealed to be associated with fat deposition and growth traits in Chinese Tan sheep [21].

The BBS have high genetic quality traits adapted to the harsh environment of low oxygen in the plateau and the lack of information on their SNPs data, this objective of this study was to (1) identify SNPs that are specific to BBS or different from other breeds, (2) infer the evolutionary history and demographic dynamics of BBS, and (3) explore the potential adaptive significance and functional consequences of these SNPs for BBS. The results of this study will not only enhance our understanding of the genetic diversity and adaptability of BBS but also provide valuable resources for future genetic improvement and conservation strategies. It is expected that the findings of this study will contribute to the genetic and genomic research of sheep and other livestock species.

## Materials and methods

### Animal and ethics approval

The experiments were carried out following the guidelines of the animal care committee of Xichang University (No. XCU20230918). The test sheep were sourced from the BBS Breeding Farm in Butuan County and genomic DNA extraction was performed on blood collected from the jugular vein (Blood was taken from the jugular vein without any anesthesia. There was no health risk to the animal.). A high-throughput DNA extraction kit was used to isolate DNA of 33 BBS from the blood. The extracted DNA underwent two types of analysis: (1) purity and integrity assessment through 1% agarose gel electrophoresis, and (2) precise quantification of DNA concentration using the Qubit system. This study subjected quantitative quality-checked DNA samples to a multiplex PCR panel mix and amplification system. The PCR reactions were carried out on a PCR instrument.

After purification of the PCR products using carboxyl magnetic beads, a second round of PCR amplification was performed using sequencing primers with attached barcodes and a high-fidelity PCR reaction system. Different barcodes were used to distinguish between various samples. The multiplex PCR capture and library construction process was completed after carboxymagnetic bead purification of the amplified products. DNA samples that passed the quality control measures underwent library construction using the appropriate products for targeted sequencing. Following library construction, preliminary quantification was performed using Qubit 2.0, followed by accurate determination of the effective concentration of the library using qPCR to ensure library quality. Once the library was successfully qualified, the samples proceeded to the next stage of high-throughput sequencing. Beijing Compass Biotechnology Co., Ltd. (http://www.kangpusen.com/) produced all the aforementioned and conducted 60K chip resequencing, and the raw data was preserved at the NCBI database (No. PRJNA1079389).

## Data analysis

### Marker loci and quality control

To avoid anomalies in population analysis that could lead to erroneous results, potentially caused by rare alleles, loci with high missing and heterozygosity rates, the original marker loci were filtered by the following parameters: loci with a minor allele frequency (MAF) of less than 0.05 were removed; loci with a missing rate of more than 20% were removed; loci with a heterozygosity ratio of more than 60% were removed; non-biallelic loci were removed.

To ensure accurate data analysis, the following filtering conditions were applied to the raw data after sequencing: remove reads with adapters attached; discard paired reads if the N content in a sequencing read exceeds 1% of the total bases in that read; exclude paired reads where the number of low-quality bases ($Q< = 5$, $Q = -10*lgP$) in a sequencing read exceeds 50% of the total bases in that read. Based on the analysis above, it is confirmed that all base quality scores Q are greater than 3, indicating an error detection rate of less than 0.001.

### Alignment

High-quality clean reads were obtained by performing quality control and then were aligned to the reference genome using the Burrows-Wheeler-Alignment Tool (BWA) software (version 0.7.17, https://sourceforge.net/projects/bio-bwa/files/). Subsequently, the sequencing depth and genome coverage of each sample were calculated. Alignment resulted in a minimum alignment rate requirement of 95%.

### Detection and annotation of SNPs

Following alignment to the sheep genome, Genome Analysis Toolkit (GATK) software (version 4.0, https://software.broadinstitute.org/gatk/) was used to detect SNP variations across the entire genome. Before analysis, the reads were aligned and removed duplicates against the reference genome. After the alignment, SNP and INDEL variants were identified using the HaplotypeCaller module, and the respective files were generated. To filter SNPs and INDELs, the VariantFiltration module, which applied strict metrics like QD (Quality by Depth), FS (Fisher Strand Bias), MQ (Mapping Quality), SORMQRankSum (Mapping Quality Rank Sum Test), and ReadPosRankSum (Read Position Rank Sum Test) were used. Finally, the SNP variations underwent strict filtering criteria. These included SNP cluster filtering, where no more than two SNPs were allowed within a 5bp region, nearby SNP filtering for INDELs (removing SNPs within 5bp of an INDEL), adjacent INDEL filtering (the distance between two INDELs

should not be less than 10bp), and any genotype with GQ (Genotype Quality) less than 20.0 was marked as lowGQ in the sample.

The SNPs were annotated using the Annovar software. Moreover, variants in the exonic region were separately annotated.

### Detection and annotation of small INDELs

INDELs in short fragments (<10 bp) also indicate differences between the sample and reference genome. INDELs in coding regions may cause frameshift mutations, resulting in gene function alterations. Following the acquisition and alignment of clean reads, the Haplotype-Caller module of the GATK tool was employed to detect INDELs. Specifically, INDELs with a depth threshold of 10x, a quality score of $\geq$30, and a mutation frequency of $\geq$0.05 were considered significant.

### Functional analysis of genes associated with SNP and INDEL variations

Enrichment analysis was conducted after identifying genes implicated in SNP and INDEL variations. The R package clusterProfiler (version 4.05) was utilized for gene function annotation and visualization. Gene Ontology annotation, which incorporates three categories: biological process (BP), molecular function (MF), and cellular component (CC), was conducted via the enrichGO function. The enrichKEGG function was implemented to annotate the potential signaling pathways in which the DEGs might be engaged, utilizing the Kyoto Encyclopedia of Genes and Genomes (KEGG). The R package ggplot2 was used to visualize all the results generated by the enrichment analysis.

### Identification of essential genes

The online tool Strings (version 11.0, available at https://string-db.org/) was used to analyze the protein-protein interactions of genes implicated in SNP and INDEL variations, revealing their interactions (parameters: Organism: *Ovis aries*; minimum required interaction score was set to high confidence 0.7, other parameters were set to default to obtain high confidence of protein network interactions). Cytoscape software (version 3.6.1) was used to visualize the interactions found. Essential genes were also identified using the Cytoscape plugin Cyto-Hubba, which identified the intersection of the top 20 DEGs ranked by four methods (Degree, EPC, MCC, MNC) as essential genes. Moreover, critical subnetworks were identified, and functional enrichment analysis was performed using the Cytoscape plugin MCODE. The resulting seed genes were also classified as essential genes.

## Results

### Quality control statistic

Over 36G of the total sequencing amount was obtained through the re-sequencing process (**Table 1**), all details are shown in (**S1 Table**). The range of raw reads averaged 164,023,208, ranging from 121,365,097 to 258,903,016. Following the elimination of low-quality sequences, the content of clean reads went from 121,353,890 to 258,896,358, with an average of 163,998,651. The base error discovery rate (Q30) averaged 95.21%, ranging from 93.63% to 96.47%. The GC content averaged 43.13%, ranging from 42.13% to 43.72%. It was observed that the resequencing results obtained were moderately accurate.

**Table 1. Sequencing information statistics of 33 BBS population.**

| sample | maximum | minimum | mean | standard deviation |
|---|---|---|---|---|
| raw reads | 258,903,016 | 121,356,097 | 164,023,208 | 36,625,189 |
| clean reads | 258,896,358 | 121,353,890 | 163,998,651 | 36,633,472 |
| Q20 (%) | 98.99 | 98.00 | 98.55 | 0.26 |
| Q30 (%) | 96.47 | 93.63 | 95.21 | 0.78 |
| GC (%) | 43.72 | 42.13 | 43.13 | 0.34 |

## Mapping statistic

Relevant statistical data can be obtained by aligning the total reads to the genome (Table 2, S2 Table). The research results indicate that the total reads range from 244,837,496 to 525,220,801, with an average of 331,174,653. The number of sequences mapped successfully to the reference genome ranges from 230,256,924 to 496,082,600, with an average of 313,704,075 —the alignment rate averages at 95.64%, with a range of 94.87% to 97.38%. The percentage of duplicated alignment reads falls between 4.91% and 8.31%, with an average of 6.08%, indicating that the rate of unique alignment positions is higher than 95%. These findings imply some variation in the total number of read sequences, mapped sequences, alignment rate, and proportion of duplicate sequences in the samples studied, but all of them lie within a reasonable range. These results have valuable reference implications for subsequent scientific research and data analysis.

## Distribution of SNP quality and type statistics

The accuracy and reliability analysis of sample SNP detection results indicate that the identified SNPs are consistent and similar across all samples. This suggests that the SNP detection in this study meets the requirements and provides a significant reference value for subsequent research and analysis (Fig 1A and 1B). Additionally, by statistically analyzing mutation types, it was found that among the existing six mutation types (C: G>A: T, C: G>G: C, C: G>T: A, T: A>A: T, T: A>C: G, and T: A>G: C), the C: G >T: A and T: A >C: G mutation types are the most frequent (>400,000 occurrences), while the T: A>A: T mutation type is the least regular (less than 200,000 occurrences) (Fig 1C). These results suggest that the C: G>T: A and T: A>C: G mutation types are dominant mutation types, which may be closely related to the evolutionary history of the BBS.

Further statistical analysis of mutation types indicates that base transition ranges from 8,874,483 to 9,716,534 with an average of 9,316,743, and base transversions range from 3,782,145 to 4,189,671 with an average of 3,985,658 (Table 3). The ratio of transitions to transversions is approximately 2.33, suggesting that base transitions happen at a rate of about 2.33 times that of transversions. Within these SNPs, heterozygotes range from 6,688,682 to 8,533,738, with an average of 7,791,796. Homozygotes range from 5,247,508 to 5,989,638, with

**Table 2. Mapping information statistics of 33 BBS population.**

| sample | maximum | minimum | mean | standard deviation |
|---|---|---|---|---|
| total reads | 525,220,801 | 244,837,496 | 331,174,653 | 74,353,947 |
| mapped reads | 496,082,600 | 230,256,924 | 313,704,075 | 70,122,732 |
| mapped ratio (%) | 97.38 | 94.87 | 95.64 | 0.50 |
| dup ratio (%) | 8.31 | 4.91 | 6.08 | 0.67 |

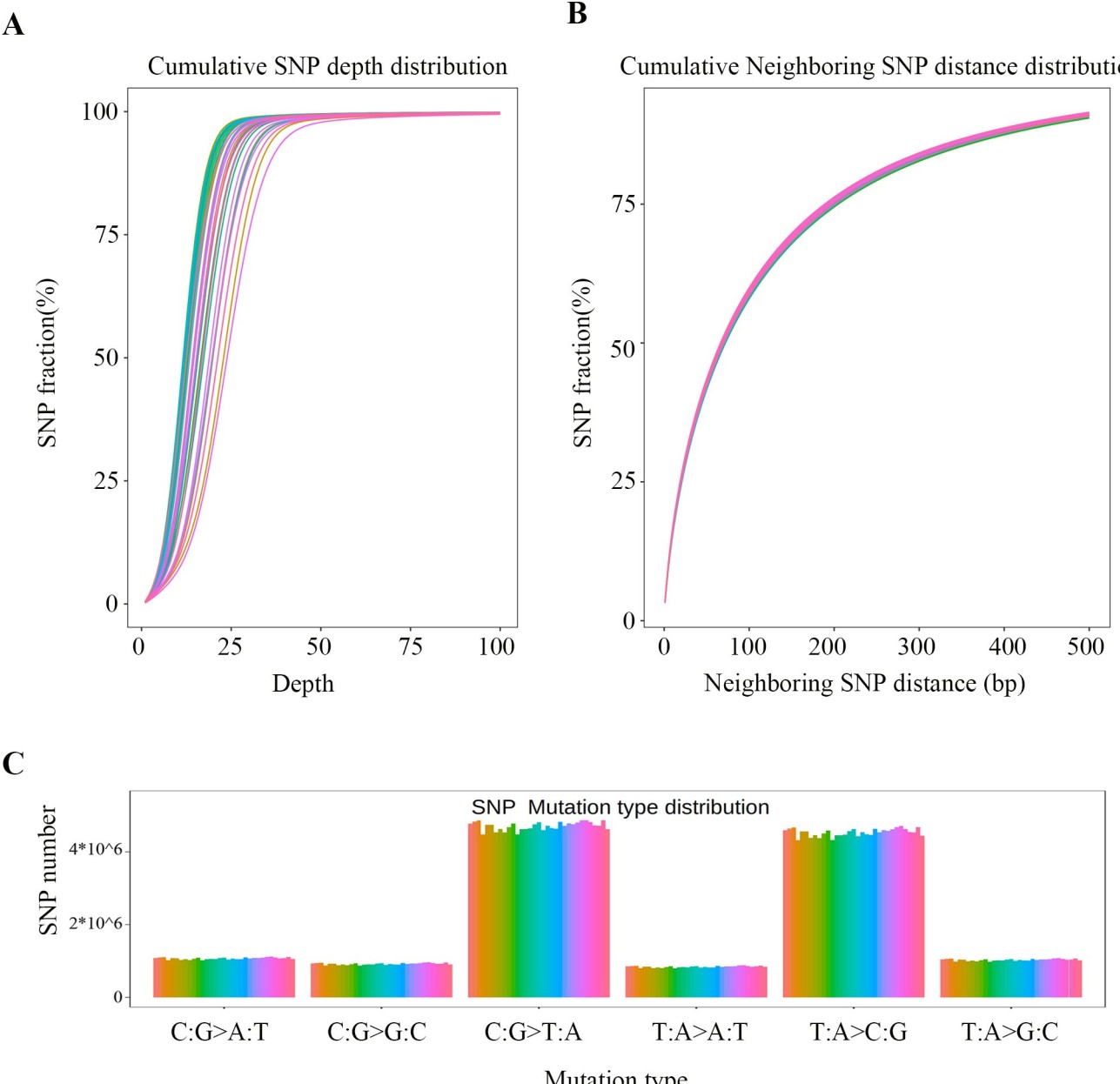

**Fig 1. Statistic of mutation information.**

**Table 3. SNP type statistics of 33 BBS population.**

|  | max | min | mean | standard deviation |
|---|---|---|---|---|
| SNPnumber | 13,906,205 | 12,659,300 | 13,302,482 | 332,415 |
| Transition | 9,716,534 | 8,874,483 | 9,316,743 | 46,251 |
| Transversion | 4,189,671 | 3,782,145 | 3,985,658 | 33,128 |
| Ti/Tv | 2.35 | 2.31 | 2.33 | 0.48 |
| Heterozygosity | 8,533,738 | 6,688,682 | 7,791,796 | 41,368 |
| Homozygosity | 5,989,638 | 5,247,508 | 5,507,476 | 39,276 |

Table 4. Variant position statistics of 33 BBS population.

| | min | max | Mean | standard deviation |
|---|---|---|---|---|
| UTR3 | 71,500 | 77,431 | 74,925 | 2,984 |
| UTR5 | 56,802 | 62,109 | 59,515 | 1,847 |
| downstream | 81,565 | 88,208 | 84,518 | 3,100 |
| upstream | 81,911 | 91,097 | 86,243 | 2,874 |
| exonic | 98,759 | 109,906 | 103,573 | 6,427 |
| intergenic | 8,022,017 | 8,856,633 | 8,440,805 | 42,145 |
| intronic | 4,431,525 | 4,860,611 | 4,642,039 | 14,126 |
| splicing | 524 | 598 | 561 | 55 |

an average of 5,507,476. These findings indicate that base transitions dominate the BBS population, with a higher probability of allele heterozygosity than homozygosity.

## Differential SNP detection and annotation

Further statistical analysis demonstrates that the range of mutations in UTR3 was 71,500 to 77,431, with a mean of 74,925 (Table 4). The content of mutations in UTR5 was from 56,802 to 62,109, with a standard of 59515. In Downstream, the modifications ranged from 81565 to 88208, with a mean of 84,518, while in Upstream, they ranged from 81,911 to 91,097, with a mean of 86243. Variation in Exonic was between 98,759 and 109,906, with a standard of 103,573. Variation in Intergenic was between 8,022,017 and 8,856,633, with a mean of 8,440,805. Variation in Intronic was between 4,431,525 and 4,860,611, with a mean of 4,642,039. Variation in Splicing was between 524 and 598, with a standard of 561. These findings highlight various SNP positions among the population of BBS, encompassing regulatory, coding, and non-coding regions. This genetic variation could give rise to the observed diversity in the BBS population.

## Identification of important mutation sites

Further analysis revealed that specific SNP mutation sites were closely associated with approximately 200 functional genes (S4 Table). These genes have been categorized into four groups based on the mutation sites: UTR (UTR 3' or UTR 5'), InterIntro (intronic and intergenic), Exonic, and downstream. Functional enrichment analysis was performed on these genes to investigate their potential functions (S5 Table). Regarding UTR genes, they were principally enriched in pathways associated with inflammation, including Cytokine-cytokine receptor interaction, Chemokine signaling pathway, Hepatitis C, Hepatitis B, immune response, and positive regulation of interleukin-1 beta production (Fig 2). The *ACAA2* and *BAD* genes are implicated in cellular response to hypoxia.

Functional enrichment analysis of the genes contained in InterIntro revealed that the mutations affect genes primarily responsible for transcriptional regulation processes (Fig 3), including gene expression, glutamatergic synapses, nuclei, transcriptional preinitiation complex, transcriptional activator activity, and RNA polymerase II transcriptional regulatory regions. This functional role is consistent with the sequence contained within this region. Additionally, two extensively researched signaling pathways have been identified: the PI3K-Akt signaling pathway (which involves genes *ITGB1, GHR, GH, NTRK2, ITGA4, ITGA11, VEGFC*) and Growth hormone synthesis, secretion, and action (which includes genes *GHR, MAPK10, GHRHR, GH*).

## Count

| chemokine activity | protein phosphatase binding | Chemokine signaling pathway | Hepatitis C | cellular response to hypoxia | | negative regulation of myoblast differentiation |
|---|---|---|---|---|---|---|
| | | | | cellular response to nicotine | | |
| Cytokine-cytokine receptor interaction | Viral protein interaction with cytokine and cytokine receptor | Hepatitis B | immune response | extrinsic apoptotic signaling pathway via death domain receptors | positive regulation of interleukin-1 beta production | positive regulation of phagocytosis |

0.00  0.01  0.02  0.03  0.04  0.05  0.06  0.07  0.08

PValue

**Fig 2. Enrichment analysis of genes related to UTR mutation.**

In **exonic** mutation-related genes, they are predominantly enriched in two functional pathways (**Fig 4**): the PI3K-Akt signaling pathway, the AMPK signaling pathway, and positive regulation of the p38MAPK cascade. These genes also play a role in signaling pathways connected to reproduction and inflammation, such as ovarian steroidogenesis and uterus development. They also impact melatonin receptor activity relating to coat color regulation (*MTNR1A*, *MTNR1B*), as well as inflammation and immune-related pathways, including hypertrophic cardiomyopathy, cytokine-cytokine receptor interaction, signaling pathways regulating pluripotency of stem cells and phospholipase C-activating G-protein-coupled receptor signaling pathway. Technical term abbreviations have been explained upon first use of the term.

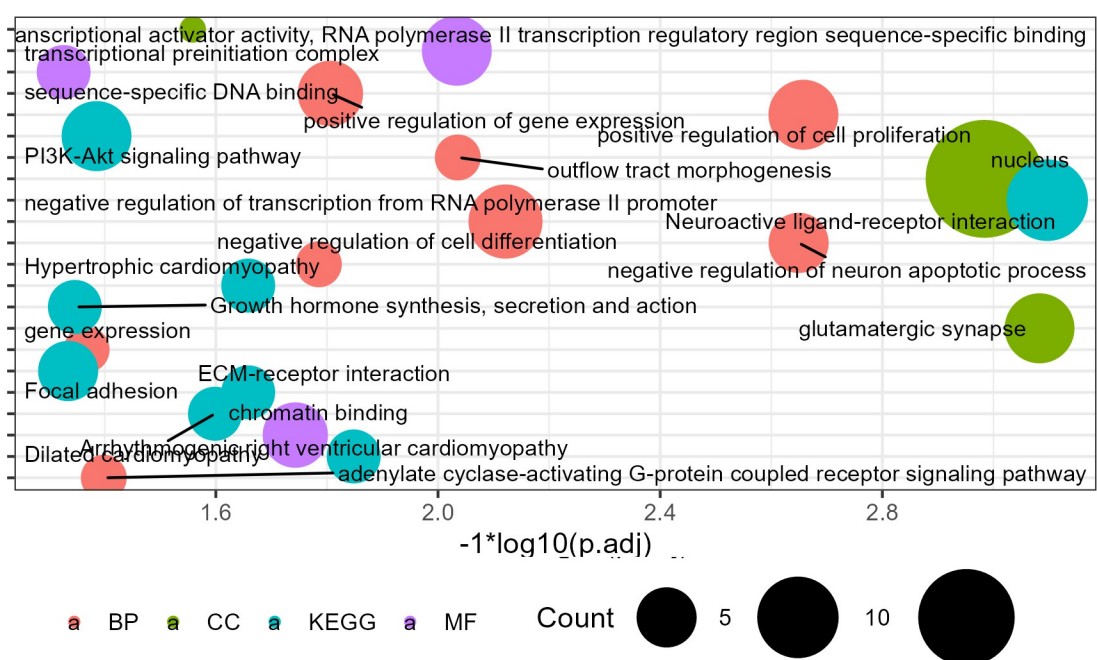

**Fig 3. Enrichment analysis of genes associated with InterIntro mutation.**

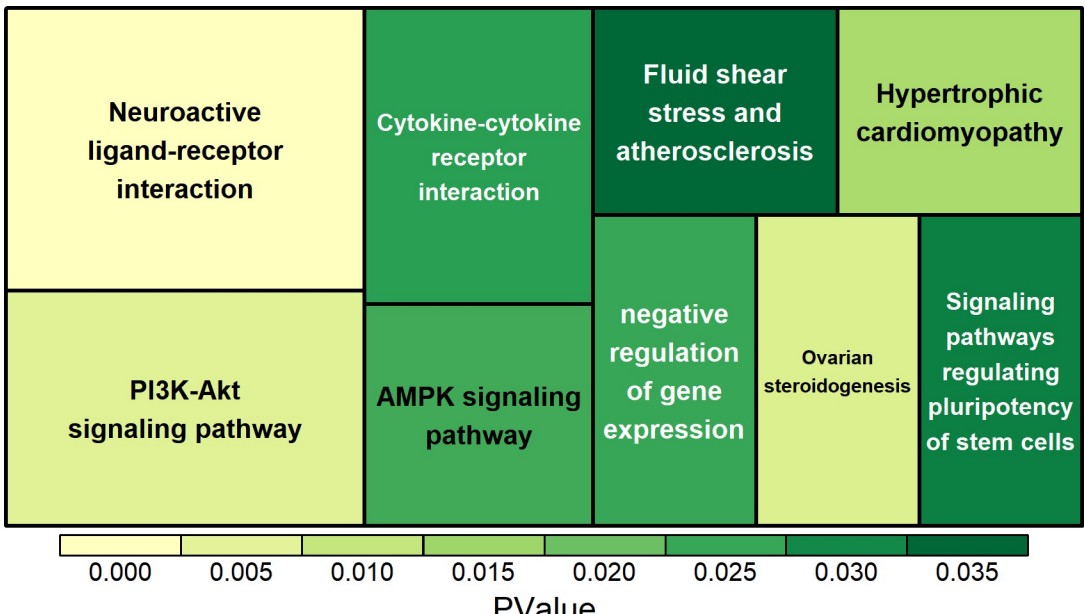

**Fig 4. Enrichment analysis of genes related to exonic mutations.**

These results suggest that the SNP variations in the population of BBS play a significant role in multiple functions. This might have been a trait that BBS developed during the process of evolution to adjust to the challenging environment of the local plateau with low oxygen.

Enrichment analysis was used to understand the function of the genes associated with these critical SNPs. The study computed and obtained 16 essential genes (**Fig 5, V-shaped**) using the CytoHubba plugin, namely *ACACA, BMPR1B, CSN1S1, CSN1S2, CSN2, DGAT1, ESR1, FSHR, FST, GHR, LACB, LALBA, LPL, PGR, PPARGC1A, STAT6*. The annotation of these core genes (**Table 5**) divulged that they could be categorized into three classes: Reproductive-related genes (*GHR, FSHR, PGR, BMPR1B, FST, ESR1*), lipid-related genes (*PPARGC1A, STAT6, DGAT1, ACACA, LPL*), and protein-related genes (*CSN2, LALBA, CSN1S1, CSN1S2*) are potentially linked to the growth and reproduction of BBS. These mutations may result in the production of single lambs and slow growth in the breed.

## Discussion

The genetic progress, growth, and reproduction of BBS pose limitations. Its genetic progress and development are relatively slow, potentially due to specific genetic mutations. Furthermore, their reproductive rate usually results in single lambs, postulating constraints to population growth. To better understand these limitations and potential genetic reasons, it is necessary to perform genomic resequencing of BBS blood. This technique will enable us to identify the distribution of SNPs, offering possible insights into the genetic factors influencing these limitations.

Specific genes associated with BBS reproduction were identified, as expected. Mutations in these genes can result in a small litter size of only one lamb, which limits BBS development. In cattle, variants of the *GHR* gene are associated with reproductive and milk production traits [22, 23]. Meanwhile, in sheep, polymorphisms in the *FSHR* gene are correlated with differences in litter size [24, 25]. Additionally, the *PGR* gene's variants play a role in reproductive traits for cattle [26]. Sheep show an increase in litter size and ovulation rate related to the

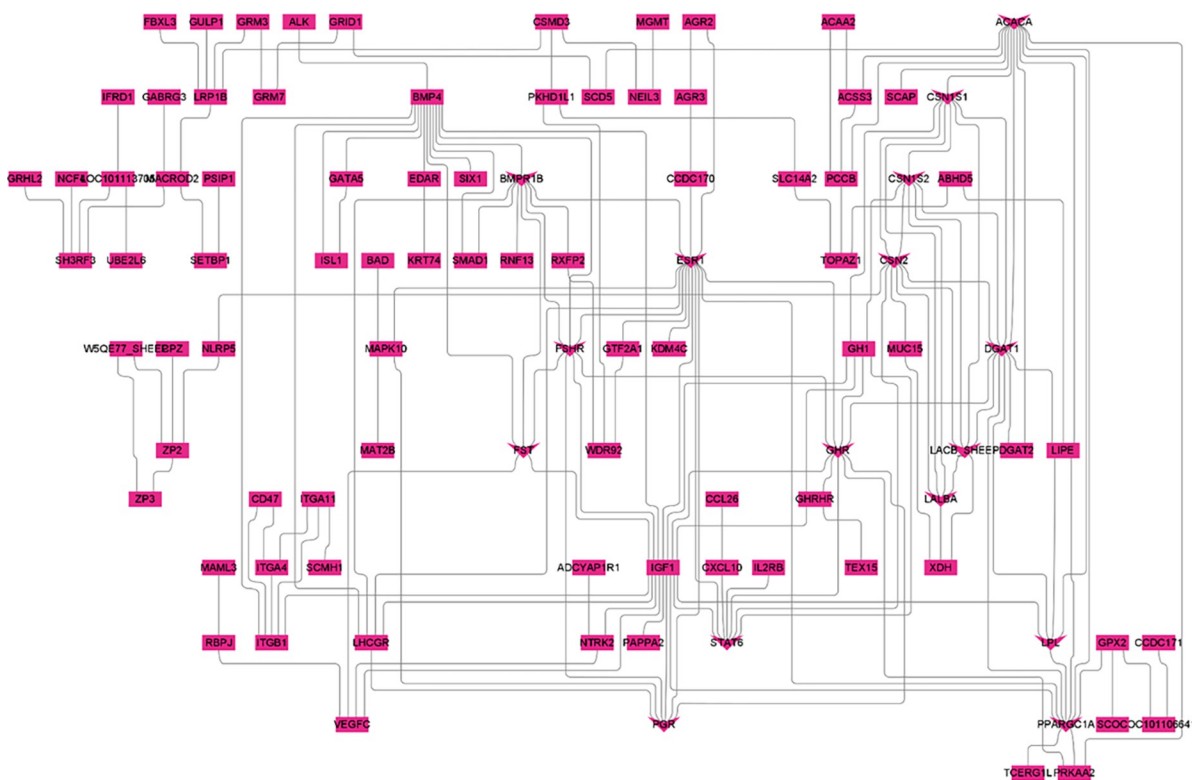

**Fig 5. PPI analysis of hub genes.**

mutation of the *BMPR1B* gene (*FecB*) [27–30]. For goats, variants of the *FST* gene are linked with litter size [31]. Lastly, polymorphisms in the *ESR1* gene have been associated with litter size in pigs and sheep [32–34]. The FecB gene regulates sheep reproduction by influencing ovulation rate and litter size among these genes. Mutations in the *FecB* gene significantly increase ovulation rate and litter size, resulting in the birth of twins or triplets [35]. Meanwhile,

**Table 5. Annotation of hub genes.**

|  | Gene Symbol | Description |
|---|---|---|
| Reproductive-related | GHR | growth hormone receptor |
|  | FSHR | follicle stimulating hormone receptor |
|  | PGR | progesterone receptor |
|  | BMPR1B | bone morphogenetic protein receptor type 1B |
|  | FST | follistatin |
|  | ESR1 | estrogen receptor 1 |
| Lipid-related | PPARGC1A | PPARG coactivator 1 alpha |
|  | STAT6 | signal transducer and activator of transcription 6 |
|  | DGAT1 | diacylglycerol O-acyltransferase 1 |
|  | ACACA | acetyl-CoA carboxylase alpha |
|  | LPL | lipoprotein lipase |
| Protein-related | CSN2 | casein beta |
|  | LALBA | lactalbumin alpha |
|  | CSN1S1 | casein alpha s1 |
|  | CSN1S2 | alpha-S2-casein |

the mutation enhances ovarian quality, increasing fertilization, and pregnancy rates [36, 37]. Additionally, two genes, *MTNR1A* and *MTNR1B*, have been identified, which are implicated in animal reproductive processes and the regulation of animal coat color. Research has demonstrated that *MTNR1A* may facilitate melatonin synthesis, whereas *MTNR1B* has the converse effect [38, 39]. By regulating the activity of tyrosine hydroxylase, *MTNR1A* and *MTNR1B* can affect the synthesis and dispersion of melatonin, ultimately influencing skin color. In addition, the level of melatonin is closely linked to the reproductive performance of Hu sheep [40–43] and varies with changes in the oestrus cycle [43, 44].

Additionally, mutations closely related to immunity and inflammation have been discovered, which could enable BBS to endure challenging environments above 3100 m in Butuo County, allowing them to grow normally. Moreover, the genes *ACAA2* and *BAD* related to mutations are involved in the cellular response to the hypoxia pathway, indicating that BBS potentially possesses a high adaptability to high-altitude hypoxia. In the hypoxic environment of high altitudes, immunity, and inflammation are regulated by various factors and signaling pathways. For instance, activating the NF-κB signaling pathway under high-altitude hypoxia aids animals in maintaining fundamental physiological functions [45]. Furthermore, the regulation of inflammatory mediators, like IL and TNF, shifts under high-altitude hypoxia, thereby boosting the animal's immune system stability [46, 47]. The *ACAA2* gene primarily involves fatty acid degradation and can abolish the cell apoptosis caused by hypoxia treatment [48]. Presently, there is no direct evidence supporting the role of this gene in promoting high-altitude hypoxia adaptation in research on animal adaptation to high-altitude hypoxia. However, the downregulation of *ACAA2* in hepatocellular carcinoma boosts adaptation to hypoxia and triggers epithelial-mesenchymal transition. It also stimulates the NF-κB signaling pathway and accelerates the loss of adipose tissue in hypoxia [49]. The *BAD* gene expression level increases 2-fold under hypoxia induction, and the methylation status of CpG islands is affected [50]. The *BAD* gene regulates cell apoptosis primarily by affecting the phosphorylation levels of Ser112, Ser136, and Ser155, participating in cell apoptosis regulations [51]. The BAD gene's expression levels elevate in heart, liver, and intestinal cells exposed to hypoxia [52].

Two general signaling pathways, PI3K-Akt and AMPK, have been identified as playing essential roles in inflammation, immunity, reproduction, growth, energy metabolism, and substance metabolism. These results are in agreement with previous findings on related genes. The PI3K-Akt signaling pathway is associated with cell proliferation, survival, and differentiation [53, 54]. It regulates the function of immune cells and inflammation responses [55, 56] and controls the development and operation of reproductive cells [57, 58]. It also regulates energy and substance metabolism [59], affecting glucose metabolism, fat metabolism, and protein synthesis. AMPK, the cellular energy sensor, can regulate cells' energy balance and metabolic pathways. It regulates energy and substance metabolism, facilitating glucose uptake and oxidation, restraining fatty acid and cholesterol synthesis, and enhancing protein synthesis and degradation [60–63]. At the same time, the AMPK signaling pathway regulates inflammatory responses and immune cell function and exerts a regulatory effect on cellular growth and proliferation [64–66]. In summary, the PI3K-Akt and AMPK signaling pathways are intricately linked and interact in various biological processes, co-regulating critical physiological functions such as inflammation, immunity, reproduction, growth, energy metabolism, and substance metabolism.

## Conclusions

In conclusion, the study offers valuable insights into BBS's genetic diversity and adaptability. Identifying significant genes and signaling pathways linked to reproduction, immunity,

inflammation, and hypoxia enhances understanding of BBS and is relevant for breeding and conservation efforts. Above all, these genes should be deem as markers in the BBS population, but more functional experiments should be conducted. These findings are significant for genetic enhancement and conservation strategies in sheep and other livestock species.

## Supporting information

**S1 Table. Statistics of sequencing results for 33 samples.**
(XLSX)

**S2 Table. Mapping statistic for 33 BBS.**
(XLSX)

**S3 Table. Variant position statistics for 33 BBS.**
(XLSX)

**S4 Table. Relevant gene information for important mutation sites for 33 BBS.**
(XLSX)

**S5 Table. Functional enrichment analysis for 200 genes.**
(XLSX)

## Author Contributions

**Conceptualization:** Chaoyun Yang.

**Data curation:** Zengwen Huang.

**Investigation:** Dongming Qi, Jinkang Wang, Yan Ruan.

**Project administration:** Jing Wang.

**Resources:** Dongming Qi, Jinkang Wang.

**Software:** Xiaoyan Li, Jingwen Zhou, Youse Laer, Zhangjia Baqian.

**Validation:** Xiaoyan Li.

**Writing – original draft:** Zengwen Huang.

**Writing – review & editing:** Chaoyun Yang.

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
