## [Decision Letter · Decision Letter 0]

19 Dec 2023

PONE-D-23-32748Uncovering the genetic diversity and adaptability of Butuo Black Sheep through whole-genome re-sequencingPLOS ONE

Dear Dr. Yang,

Thank you for submitting your manuscript to PLOS ONE. After careful consideration, we feel that it has merit but does not fully meet PLOS ONE’s publication criteria as it currently stands. Therefore, we invite you to submit a revised version of the manuscript that addresses the points raised during the review process.

**ACADEMIC EDITOR: Kindly attend to every comment raised by the Reviewers and respond accordingly as quickly as possible.**

We look forward to receiving your revised manuscript.

Kind regards,

Olatunji Matthew Kolawole, Ph.D.

Academic Editor

PLOS ONE

Journal Requirements:

2. We noted that animal blood samples were used in your study but no details of the animal sources were provided. To comply with our policy on animal research submission (https://journals.plos.org/plosone/s/submission-guidelines#loc-animal-research), please specify the source of blood samples used used.

Thank you for your attention to the additional requests.

"The authors declare that they have no competing interests."

5. In the online submission form, you indicated that [We will provide the data if requested.]. 

6. Please amend your list of authors on the manuscript to ensure that each author is linked to an affiliation. Authors’ affiliations should reflect the institution where the work was done (if authors moved subsequently, you can also list the new affiliation stating “current affiliation:….” as necessary).

7. Please ensure that you refer to Figure 2 in your text as, if accepted, production will need this reference to link the reader to the figure.

Reviewers' comments:

Reviewer's Responses to Questions

**Comments to the Author**

1. Is the manuscript technically sound, and do the data support the conclusions?

Reviewer #1: Yes

Reviewer #2: Yes

2. Has the statistical analysis been performed appropriately and rigorously? 

Reviewer #1: No

Reviewer #2: Yes

3. Have the authors made all data underlying the findings in their manuscript fully available?

Reviewer #1: Yes

Reviewer #2: Yes

4. Is the manuscript presented in an intelligible fashion and written in standard English?

Reviewer #1: Yes

Reviewer #2: Yes

5. Review Comments to the Author

Reviewer #1: This study aimed at identifying SNPs that are specific to BBS or different from other breeds; inferring the evolutionary history and demographic dynamics of BBS and; exploring the potential adaptive significance and functional consequences of these SNPs for BBS.

The reviewer thinks the manuscript is good and will be relevant to conservation and breeding improvement efforts.

The following points may be considered to improve the manuscript

The last sentence on the introduction is not adding any information to the manuscript. Please consider removing it.

Give more details of the material and method. E.g. no information on the specific BBS used in the study. At least a brief details of the DNA extraction and resequencing shouybe provided.

What's the justification for your quality control and filtering parameters through out the manuscript? I think you should provide some references.

L127-129: "The online tool Strings (version 11.0, available at https://string-db.org/) was used to analyze

the protein-protein interactions of genes implicated in SNP and INDEL variations, revealing their

interactions." Can you give more details and references on this? Did you apply any statistics?

L14-143: It was observed that the resequencing results obtained were moderately accurate. Can you provide some references to this?

The table names are unacceptable. Use for example: Table 1. Sequencing information statistics of 33 BBS population.

The table can also have the highest and the lowest five samples followed by the mean, minimum and maximum.

L151-153: what is the measurable range? Who determines this? I think statements like this and several others in the results section explaining the implications of the results should come in discussion section. No?

L154: expand more on the valuable reference implications.

L212: Technical term abbreviations have been explained upon first use of the term. Do you really need to have this here?

Did you perform genomic resequencing of BBS using blood samples or genomic resequencing of BBS blood? This should be properly presented throughout the MS.

No reference to Table S1 in the text.

Reviewer #2: Comments

The manuscript draft entitled "Uncovering the genetic diversity and adaptability of Butuo Black Sheep through whole genome re-sequencing " has important contents worth communicating to the scientific community. The study seems scientifically sound, and the results are presented in a clear and concise manner. However, there are a few points that need to be addressed.

Abstract:

The abstract start with the aim of the study and then the background, but it is better to present the background of the study followed by objectives.

Line 15= SNPs (abbreviation along with full form should be written when it appears for the first time

Lack of clarity between the significance of the study and recommendation.

Line24-25 seem significant of the study, and the author should make appropriate recommendation

Keywords

Keywords should be presented in alphabetical order

Make the keyword “Black sheep” to “Buton black sheep”

Whole genome resequencing should be a keyword

Introduction

Line 28, make “Ovis aries” italic

There are spacing problems on the reference citation

Lines 50 and 52, “Single nucleotide polymorphisms (SNPs)” redundancy

Lines 61 and 62 should be present in the material and methods part

Lacks research question

The objectives should come after the research question

Material and Methods

It lacks some clarity such as study setting, study population, sample collection method, sample processing including DNA extraction

Line 72, All tissues? Which??? Makes confusion

The authors claimed that “the experiments were carried out following the guidelines of the animal care committee of Xichang University”, but this is not enough and ethical clearance letter should be obtained from the relevant ethical committee and the reference number of the approved letter should include.

Line 84-85, spacing problem

Line 91, full form of BWA required

Line 96, full form of GATK required

Line 99, full form of INDEL required

Results

The authors presented the minimum, maximum and mean of each read, Hence, if there is a mean, standard deviation and/or standard error should be presented along with it.

The Authors present simply figures such as 164,023,208. To make it clearly show the reads were nucleotides

Line 170, again “Single nucleotide polymorphisms (SNPs)” repeated, the author should write the abbreviated form only when it appears for the second and more times

Discussion

Needs some modification

Spacing of reference citations

Line 228-231 looks like justification of the study and it is better to write in the introduction part

Better to use passive voice instead of saying “we have identified specific genes” line 232. “we have identified” line 271 etc.

What is the relevance of writing sentences like “In cattle, variants of the GHR gene are associated with reproductive and milk production traits” line 234 or “the PGR gene's variants play a role in reproductive traits for cattle” line 236??? Instead, write the different breeds of sheep that have similar or different genes responsible for the variation or similarity.

Line 283-286, repeated sentence of line 274-275

Conclusions:

Lacks the recommendations

References

Make Reference to References

6. PLOS authors have the option to publish the peer review history of their article (what does this mean?). If published, this will include your full peer review and any attached files.

Reviewer #1: **Yes: **Kehinde Adewole Adeboye, Ekiti State Polytechnic, Isan, Nigeria

Reviewer #2: No

---

## [Author Response · Author response to Decision Letter 0]

11 Apr 2024

Response to Reviewers

Dear Editor and Reviewers, 

we appreciate the time and effort you have dedicated to reviewing our manuscript. This revision aims to meet the publication requirements of your journal by addressing formatting issues according to the journal style requirements. Firstly, we have addressed all the queries of the editors and reviewers. Secondly, we have uploaded the raw data to the NCBI database under the number PRJNA1079389. Details of the revision are given below:

Editor :

Q1: We note that the grant information you provided in the ‘Funding Information’ and ‘Financial Disclosure’ sections do not match. 

Response: We have checked that it is correct.

Q2: Thank you for stating the following in your Competing Interests section: 

"The authors declare that they have no competing interests."

Response: The manuscript now includes the Competing Interests section.

Response: This information has been added.

Q3: In the online submission form, you indicated that [We will provide the data if requested.]. 

Response: The raw data of 33 BBSs has been uploaded to the SRA database (project number PRJNA1079389).

Reviewers' comments:

5. Review Comments to the Author

Reviewer #1:

 This study aimed at identifying SNPs that are specific to BBS or different from other breeds; inferring the evolutionary history and demographic dynamics of BBS and; exploring the potential adaptive significance and functional consequences of these SNPs for BBS.

The reviewer thinks the manuscript is good and will be relevant to conservation and breeding improvement efforts.

The following points may be considered to improve the manuscript

Q1: The last sentence on the introduction is not adding any information to the manuscript. Please consider removing it.

Response: The sentence have been removed.

Q2: Give more details of the material and method. E.g. no information on the specific BBS used in the study. At least a brief details of the DNA extraction and resequencing should be provided.

Response: Steps related to DNA extraction and sequencing have been added. Please review the modified version.

Q3: What's the justification for your quality control and filtering parameters through-out the manuscript? I think you should provide some references.

Response:The software parameters used were higher than the default values. Citations to appropriate references have been added to the manuscript. Please refer to the revised manuscript.

Q4: L127-129: "The online tool Strings (version 11.0, available at https://string-db.org/) was used to analyze the protein-protein interactions of genes implicated in SNP and INDEL variations, revealing their interactions." Can you give more details and references on this? Did you apply any statistics?

Response: Please refer to the revised version for additional details and parameters regarding the use of this site. The technical interactions are automatically calculated by the system and have not been modified.

Q5: L14-143: It was observed that the resequencing results obtained were moderately accurate. Can you provide some references to this?

Response: The false discovery rate (FDR) was used to evaluate the sequencing results in our study. The percentage of reads above Q20 and Q30 were 98.99% and 96.47%, respectively (Table 1). This indicates that the probability of these reads being inappropriate is P ≤ 0.01, which is considered a small probability event. Therefore, we state in the paper that the resequencing results obtained were moderately accurate.

Q6:The table names are unacceptable. Use for example: Table 1. Sequencing information statistics of 33 BBS population.

Response:The table name for Table1-4 has been changed. Please review the revision.

Q7:The table can also have the highest and the lowest five samples followed by the mean, minimum and maximum.

Response: Based on your and the second reviewer's comments, table1-4 has been reorganized and its standard deviation has been reported, see table1-4 in the revised manuscript.

Q8:L151-153: what is the measurable range? Who determines this? I think statements like this and several others in the results section explaining the implications of the results should come in discussion section. No?

Response: Some modifications have been made in the Results section, please see the revised manuscript. Additionally, the biological phenomena that may be manifested based on the results obtained have also been described in the Results section and discussed in detail by listing the arguments in the Discussion.

Q9:L154: expand more on the valuable reference implications.

Response: The article has been thoroughly and extensively revised, please review the revised version.

Q10:L212: Technical term abbreviations have been explained upon first use of the term. Do you really need to have this here?

Response:We apologize for any confusion caused by our description. This is the first time that UTR and InterIntro have been mentioned in this context, and so we wrote the full name.

Q11:No reference to Table S1 in the text.

Response: To present comprehensive information, we have included Table 1 which displays the maximum value, minimum value, average value, and other relevant data. For detailed statistical information on all the 33 BBSs, please refer to Table S1. 

Reviewer #2: Comments

The manuscript draft entitled "Uncovering the genetic diversity and adaptability of Butuo Black Sheep through whole genome re-sequencing " has important contents worth communicating to the scientific community. The study seems scientifically sound, and the results are presented in a clear and concise manner. However, there are a few points that need to be addressed.

Abstract:

Q1:The abstract start with the aim of the study and then the background, but it is better to present the background of the study followed by objectives.

Response: The abstract has been restructured according to Background of the study → Objectives of the study → Methodology of the study → Findings and conclusions, please review the revised version.

Q2:Line 15= SNPs (abbreviation along with full form should be written when it appears for the first time 

Response: The revised manuscript includes the full names of the SNPs.

Q3: Lack of clarity between the significance of the study and recommendation.

Response: The content of the abstract has been reviewed and changes have been made; please see the revised draft.

Q4:Line24-25 seem significant of the study, and the author should make appropriate recommendation

Response: The abstract has been revised and recommendations have been made. Please review the updated version.

Keywords

Q5:Keywords should be presented in alphabetical order

Response: The order of the keywords has been rearranged alphabetically. Please review the revisions.

Q6:Make the keyword “Black sheep” to “Butuo black sheep” 

Response: We changed 'Black sheep' to 'Butuo black sheep'. Please review the revised version.

Q7:Whole genome resequencing should be a keyword

Response: We have set “Whole genome resequencing” as the keyword, please check the revised version.

Introduction

Q8:Line 28, make “Ovis aries” italic

Response: We have changed "Ovis aries" to italics, please revise the revised version.

Q9:There are spacing problems on the reference citation

Response: The spacing of all reference numbers and 3's has been standardised. Please review the revised version for accuracy.

Q10:Lines 50 and 52, “Single nucleotide polymorphisms (SNPs)” redundancy

Response: We have removed the redundant phrase 'Single nucleotide polymorphisms (SNPs)'. Please review the revised version.

Q11:Lines 61 and 62 should be present in the material and methods part

Response: We have revised the sentence in the Materials and Methods section to include the research methodology. Please review the revised manuscript.

Q12:Lacks research question

The objectives should come after the research question

Response: We have formulated research questions. Please view revised version.

Material and Methods

Q13:It lacks some clarity such as study setting, study population, sample collection method, sample processing including DNA extraction

Response: The revised manuscript now includes information on the sample collection process, as well as the related processes of DNA extraction and sequencing.

Q14:Line 72, All tissues? Which??? Makes confusion

Response: We apologize for any misrepresentation or misinterpretation. The study used only blood, not tissue, as clarified in the revised version.

Q15: The authors claimed that “the experiments were carried out following the guidelines of the animal care committee of Xichang University”, but this is not enough and ethical clearance letter should be obtained from the relevant ethical committee and the reference number of the approved letter should include.

Response:The revised version now includes the approval number of the ethics committee.

Q16: Line 84-85, spacing problem

Response:The revisions have been made, please review them.

Q17:Line 91, full form of BWA required

Response:The revised version now includes the full name.

Q18:Line 96, full form of GATK required

Response:The revised version now includes the full name.

Q19:Line 99, full form of INDEL required

Response:The revised version now includes the full name.

Results

Q20:The authors presented the minimum, maximum and mean of each read, Hence, if there is a mean, standard deviation and/or standard error should be presented along with it.

Response:Tables 2-4 have been reclassified so that they all include the maximum, minimum and mean standard deviation of the chickens; please review the revised version.

Q21:The Authors present simply figures such as 164,023,208. To make it clearly show the reads were nucleotides

Response: The article aimed to resequence the genome of blood DNA, resulting in the acquisition of nucleotides.

Q22:Line 170, again “Single nucleotide polymorphisms (SNPs)” repeated, the author should write the abbreviated form only when it appears for the second and more times

Response: The revised version has removed numerous duplicates. Please review it.

Discussion

Q23:Needs some modification

Spacing of reference citations

Response: We have removed multiple spaces in the text. Please check the revised version.

Q24: Line 228-231 looks like justification of the study and it is better to write in the introduction part

Response: The text has been revised to improve its logical structure. Please review the revised version.

Q25:Better to use passive voice instead of saying “we have identified specific genes” line 232. “we have identified” line 271 etc.

Response: We have reviewed the entire text and ensured consistent use of the passive voice. Please review the revised version.

Q26:What is the relevance of writing sentences like “In cattle, variants of the GHR gene are associated with reproductive and milk production traits” line 234 or “the PGR gene's variants play a role in reproductive traits for cattle” line 236??? Instead, write the different breeds of sheep that have similar or different genes responsible for the variation or similarity.

Response: In discussions about genes, findings from cattle are sometimes cited for two main reasons: A) The gene has not been found in sheep or goats. B) Both breeds are ruminants and functionally similar. Therefore, we have chosen to use the findings of this gene in beef cattle to support our own findings.

Q27: Line 283-286, repeated sentence of line 274-275

Response:The article was re-checked and changes were made.

Conclusions:

Q28:Lacks the recommendations

Response: Appropriate recommendations have been provided in the Conclusion section. Please review the revised version.

References

Q29: Make Reference to References

Response: All reference numbers and reference lists in the manuscript were checked to ensure correct citation. Please review the revised manuscript.

---

## [Editor Report · Decision Letter 1]

25 Apr 2024

Uncovering the genetic diversity and adaptability of Butuo Black Sheep through whole-genome re-sequencing

PONE-D-23-32748R1

Dear Dr. Yang,

We’re pleased to inform you that your manuscript has been judged scientifically suitable for publication and will be formally accepted for publication once it meets all outstanding technical requirements.

Kind regards,

Olatunji Matthew Kolawole, Ph.D.

Academic Editor

PLOS ONE
---

## [Editor Report · Acceptance letter]

30 May 2024

PONE-D-23-32748R1 

PLOS ONE

Dear Dr. Yang, 

I'm pleased to inform you that your manuscript has been deemed suitable for publication in PLOS ONE. Congratulations! Your manuscript is now being handed over to our production team.

Kind regards, 

on behalf of

Dr. Olatunji Matthew Kolawole 

Academic Editor

PLOS ONE